# Adherence to Hydroxyurea Therapy for Pediatric Sickle Cell Anemia in Tanzania: Evidence from Bugando Medical Centre

**DOI:** 10.3390/ijerph22040616

**Published:** 2025-04-15

**Authors:** Maria Inviolata Subira, Emmanuela E. Ambrose, Eveline Konje

**Affiliations:** 1School of Public Health, Catholic University of Health and Allied Sciences, Mwanza P.O. Box 1464, Tanzania; inviolatasobhera@gmail.com; 2Department of Pediatrics and Child Health, Catholic University of Health and Allied Sciences, Mwanza P.O. Box 1464, Tanzania; 3Department of Pediatrics and Child Health, Bugando Medical Centre, Mwanza P.O. Box 1370, Tanzania; 4Department of Epidemiology and Biostatistics, Catholic University of Health and Allied Sciences, Mwanza P.O. Box 1464, Tanzania; ekonje28@yahoo.com

**Keywords:** adherence, hydroxyurea, sickle cell anaemia, Tanzania

## Abstract

Hydroxyurea is effective in reducing the severity of Sickle cell anemia (SCA) symptoms, yet adherence remains challenging, particularly in resource-limited settings. Bugando Medical Centre, a major healthcare provider, faces undocumented adherence issues among its pediatric SCA patients. This study aims to evaluate the adherence rate to hydroxyurea therapy among caregivers of children with SCA at Bugando Medical Centre and identify factors contributing to non-adherence. This analytical cross-sectional study involved 172 participants. Data were analyzed using Stata version 15 and modified Poisson regression determined the association between exposures and adherence to hydroxyurea treatment. More than half (68.6%) of the children were aged between 1 and 10 years, with a median age of 8 years (IQR: 5–12). Good adherence to hydroxyurea was observed in 23.8% of participants, while 76.2% showed moderate to poor adherence. Children aged 1–10 years were twice as likely to have good adherence compared to those aged 11–17 years (aPR = 2.98, 95% CI = 1.18, 7.47). Children of caregivers with secondary education had a 41% higher chance of good adherence (aPR = 1.41, 95% CI = 1.19, 2.87) compared to those with primary education. Additionally, children of caregivers with college/university education had a 92% higher chance of good adherence (aPR = 1.92, 95% CI = 1.09, 4.63) compared to those with primary education. Participants with good knowledge of hydroxyurea had a 55% higher chance of good adherence (aPR = 1.55, 95% CI = 1.10, 4.78) compared to those with poor knowledge. Factors such as the child’s age and caregiver’s educational level are associated with good adherence to hydroxyurea treatment. Despite these associations, overall adherence rates are low, highlighting the need for targeted interventions to enhance knowledge and awareness about the importance of adherence to hydroxyurea treatment.

## 1. Background

Sickle cell anemia (SCA) affects around 300,000 children worldwide each year [1]. In Sub-Saharan Africa (SSA), where more than 75% of individuals with SCA reside, this percentage is expected to reach 85% by 2050 [1]. Tanzania ranks fifth globally, with approximately 14,000 infants born annually with SCA [2,3]. The high mortality rate in SSA, compared to high-income countries, adds to the prevalence of SCA [4]. SCA contributes significantly to under-five mortality in Tanzania, constituting 7% of all-cause deaths in this age group [5]. Sickle cell trait and disease are prevalent in newborns in northwest Tanzania, with rates exceeding 20% and 1.2%, respectively. The Lake Zone regions are predominantly affected due to low immigration and similar environmental exposures [3].

Hydroxyurea (HU) is a myelosuppressive agent that induces hemoglobin F (HbF) production, showing promising clinical benefits even before significant HbF elevation [6]. Research indicates that HU reduces the frequency and severity of acute pain crises and acute chest syndrome in SCA patients. Additionally, it decreases SCA-related mortality and the average number of hospital days by 40% [6,7]. However, insufficient knowledge about hydroxyurea, its mode of action, and its benefits can hinder patient adherence. Hydroxyurea is recognized by European and United States national institutes as the standard of care for SCA due to its potential to extend and improve the quality of life for affected children [8]. The dosage and adherence levels significantly impact the effectiveness of HU treatment. A recent study in Northwestern Tanzania found that HU at a maximum tolerated dose significantly reduces stroke occurrence in the SCA population [9].

In low- and middle-income countries, including Tanzania, the acceptance and adherence to hydroxyurea treatment remain notably low despite its proven benefits and long-standing safety record in regions like Europe, America, and Jamaica. In Tanzania, socioeconomic challenges such as high rates of poverty, limited health insurance coverage, and out-of-pocket medication costs hinder consistent access to hydroxyurea. Additionally, logistical issues such as intermittent drug stock-outs and inadequate laboratory infrastructure for monitoring therapy further impede adherence. Compared to countries like Nigeria and Uganda, Tanzania faces similar systemic barriers, although some regions have implemented more robust sickle cell programs and community awareness initiatives. The primary obstacles to adherence in Tanzania include forgetfulness, fear of side effects, low caregiver health literacy, and the burden of regular clinic visits for drug refills and monitoring [10,11,12]. A study in Nigeria highlighted the need for provider and health system-level interventions to improve hydroxyurea uptake and adherence among SCA patients [11]. These insights emphasize the importance of increasing awareness, addressing affordability and availability, and enhancing caregiver education to improve hydroxyurea adherence in Tanzania and similar contexts.

### 1.1. Problem Statement

The low uptake and poor adherence to hydroxyurea among children with sickle cell anemia (SCA) in Tanzania is a significant public health concern, particularly given hydroxyurea’s proven benefits in reducing sickle cell crises, hospitalization rates and mortality [8]. Sub-Saharan Africa accounts for over 75% of the global SCA burden, a figure expected to increase to 85% by 2050 [13]. In Tanzania, SCA contributes to approximately 7% of under-five mortality [14].

Bugando Medical Centre (BMC), a tertiary referral and teaching hospital located in Mwanza, plays a central role in managing SCA in the Lake Zone region, serving over 800 SCA patients annually through its specialized pediatric sickle cell clinic. Of these, approximately 300 children are on hydroxyurea therapy. Despite its clinical importance, there has been no prior published research assessing hydroxyurea adherence among this population at BMC. This is a critical gap, as understanding adherence patterns in this context can inform targeted interventions, especially in regions with resource limitations, fragmented follow-up systems, and variable access to medication.

In Tanzania, socioeconomic constraints such as poverty, out-of-pocket treatment costs, and limited insurance coverage—coupled with health system barriers like stock-outs and inadequate caregiver counseling—significantly hinder adherence. Additional factors, including caregiver misconceptions about drug safety and the burden of frequent clinic visits for monitoring, further contribute to poor uptake. Although studies in other Sub-Saharan countries (e.g., Nigeria) have explored these barriers, localized evidence is essential for effective policy and clinical decision-making. This study addresses that gap by investigating adherence levels and their associated factors at BMC, ultimately aiming to enhance program design and patient outcomes at the national level.

### 1.2. Rationale

This study aims to determine the adherence to hydroxyurea treatment among children with sickle cell anemia attending the sickle cell clinic at Bugando Medical Centre in Tanzania. This is to address the existing knowledge gap and improve the treatment outcomes for sickle cell anemia. Understanding these factors is essential for developing effective hydroxyurea intervention programs that can reduce sickle cell anemia-related mortality and enhance the quality of life for affected individuals.

### 1.3. Research Question

What are the factors associated with adherence to hydroxyurea treatment among caregivers of children with sickle cell anemia attending the sickle cell clinic at Bugando Medical Centre, Mwanza, Tanzania?

### 1.4. Broad Objective

To determine the adherence level to hydroxyurea treatment among caregivers of children with sickle cell anemia receiving medical care at Bugando Medical Centre.

### 1.5. Specific Objectives

1.To assess the level of adherence to hydroxyurea among children with sickle cell anemia attending the sickle cell clinic at Bugando Medical Centre.2.To identify factors associated with adherence to hydroxyurea treatment among caregivers of children with sickle cell anemia attending the sickle cell clinic at Bugando Medical Centre.

### 1.6. Study Setting

The study was conducted at Bugando Medical Centre (BMC) in Mwanza, Tanzania, a teaching and consultant tertiary referral hospital serving the Lake Zone. BMC’s catchment area includes the regions of Kagera, Geita, Shinyanga, Mwanza, Mara, and Simiyu, covering approximately 14 million people. The pediatric unit at BMC has 121 beds and operates a sickle cell clinic two days a week, treating around 800 SCA patients annually, with 300 of them receiving hydroxyurea therapy. Before starting hydroxyurea, caregivers received comprehensive information and education about the drug’s use and potential side effects.

### 1.7. Study Design

This study utilized a cross-sectional design to collect data on hydroxyurea adherence using a medication adherence questionnaire among caregivers of children with SCA attending the sickle cell clinic at Bugando Medical Centre. *p*-value < 0.05 was considered statistically significant

### 1.8. Study Population

The study population included all caregivers of children under 18 years with sickle cell anemia on hydroxyurea 20 mg/kg/day treatment, attending the sickle cell clinic at Bugando Medical Centre. Adherence was categorized based on caregiver responses using the validated adherence questionnaire: Good adherence—taking ≥80% of prescribed doses; Moderate adherence—50–79%; Poor adherence—<50%. Knowledge of hydroxyurea was assessed using a structured questionnaire with multiple-choice items. Scores were categorized as follows: Good knowledge—≥80% correct responses; Moderate knowledge—50–79%; Poor knowledge—<50%.

### 1.9. Inclusion Criteria

Caregivers of children who were under 18 years with confirmed sickle cell anemia, on hydroxyurea treatment for at least three months for adequate hydroxyurea utilization and attending the sickle cell clinic at BMC were included. Participation was voluntary; only children whom caregivers consented were included in the study.

### 1.10. Exclusion Criteria

Caregivers unwilling to share information, those with severely ill children currently experiencing a crisis, and caregivers of children who had been on hydroxyurea treatment for less than three months were excluded. Additionally, participants who missed their appointments during data collection were not included in the study.

### 1.11. Sampling Procedure

A list of patients’ names from the sickle cell register was used to conduct a simple random sampling of patients under 18 years on hydroxyurea therapy. Patients aged 18 and above or not on hydroxyurea were excluded. If a selected patient missed their appointment or was severely ill, the next available patient on the list was selected. This process continued for the entire data collection period.

### 1.12. Ethical Considerations

Ethical approval was obtained from the CUHAS and BMC joint Ethics and Review Committee. Permission to conduct the study was secured from relevant Bugando Medical Centre authorities, including the Director General, the Director of Medical Services, and the Head of the Pediatric Department. Written informed consent/assent was obtained from each participant before the interview. Participant confidentiality was maintained by using identification numbers instead of names, with information securely stored and access limited to the investigators only.

### 1.13. Dissemination of Results

The results of this study will be presented as part of the requirements for a Master’s degree in Public Health at the Catholic University of Health and Allied Sciences (CUHAS). Copies will be distributed to the CUHAS library and Public Health department, following the Tanzania Commission for University requirements for manuscript preparation. Findings will be published in a peer-reviewed journal.

## 2. Results

### 2.1. Background Characteristics of the Study Participants

The characteristics of the participants are shown in Table 1. A total of 172 participants were analyzed. More than half (68.6%) of the children were aged between 1 and 10 years. Their median age was 8 (IQR: 5–12) years. About 38.4% of the caregivers had completed secondary education, and 68.6% of them were not employed. Moreover, a large percentage (73.3%) of the caregivers had good knowledge of hydroxyurea, and 89.5% of the caregivers reported that they believe hydroxyurea treatment helps to reduce pain in their children (Table 1).

### 2.2. Adherence to Hydroxyurea Treatment Among Study Participants

The information on the adherence to hydroxyurea treatment among study participants is shown in Figure 1. A significant percentage of the participants (23.8%) had good adherence to hydroxyurea treatment, and 76.2% of the participants had moderate-to-poor adherence to hydroxyurea treatment (Figure 1).

### 2.3. Unadjusted Analysis on the Adherence to Hydroxyurea Treatment and the Associated Factors Among Study Participants

In the crude analysis, child age, education of caregivers, and knowledge of hydroxyurea were significantly associated with adherence to hydroxyurea treatment. Children who were aged 1–10 years had significantly three times more chance to have good adherence to hydroxyurea treatment (cPR = 3.29, 95% CI = 1.36, 7.95) as compared to their reference group aged 11–17 years; children who belonged to caregivers with college/university education had significantly two times higher chance of good adherence to hydroxyurea treatment (cPR = 2.26, 95% CI = 1.08, 4.70) as compared to those children belong to caregivers with primary education. However, participants with good knowledge of hydroxyurea had a significantly 96% higher chance of having good adherence to hydroxyurea treatment (cPR = 1.96, 95% CI = 1.52, 7.40) as compared to their control with poor knowledge of hydroxyurea treatment (Table 2).

### 2.4. Adjusted Analysis on the Adherence to Hydroxyurea Treatment and the Associated Factors Among Study Participants

In the adjusted analysis, caregiver age, child age, caregiver education, knowledge about hydroxyurea, and the consistent availability of hydroxyurea at the clinic were significantly associated with adherence to hydroxyurea treatment. Children aged 1–10 years had a significantly higher likelihood of good adherence (aPR = 2.98, 95% CI = 1.18, 7.47) compared to those aged 11–17 years. Children of caregivers with secondary education had a 41% higher chance of good adherence (aPR = 1.41, 95% CI = 1.19, 2.87) compared to those with primary education. Similarly, children of caregivers with college or university education had a 92% higher chance of good adherence (aPR = 1.92, 95% CI = 1.09, 4.63) compared to those with primary education. Participants with good knowledge of hydroxyurea had a 55% higher chance of good adherence (aPR = 1.55, 95% CI = 1.10, 4.78) compared to those with poor knowledge. Those with moderate knowledge had a 44% higher chance of good adherence (aPR = 1.44, 95% CI = 1.04, 4.66) compared to those with poor knowledge. Additionally, participants who reported consistent availability of hydroxyurea at the clinic had a 49% higher chance of good adherence (aPR = 1.49, 95% CI = 1.03, 3.52) compared to those who reported its absence (Table 3).

## 3. Discussion

The study included 172 participants, with 68.6% of children aged 1–10 years—a typical age range for the onset of sickle cell anemia symptoms. This finding is consistent with observations by Brouseau et al. [15]. Approximately 38.4% of caregivers had attained secondary education, a factor known to influence healthcare decisions and treatment adherence [16]. Moreover, 68.6% of caregivers were unemployed, potentially limiting their ability to effectively manage their child’s healthcare needs, as similarly reported in a study on penicillin adherence by Cober et al. [17]. Encouragingly, a substantial proportion (73.3%) of caregivers demonstrated good knowledge of hydroxyurea, which is essential for effective medication management and adherence [17]. Notably, 89.5% of caregivers believed that hydroxyurea reduced pain in their children—an outcome consistent with its established therapeutic benefits, as also reported in a multicenter study by Charache et al. [18].

### 3.1. Adherence to Hydroxyurea

The study found that only 23.8% of participants demonstrated good adherence to hydroxyurea, while the majority (76.2%) had moderate to poor adherence. Several interrelated factors contribute to this low adherence rate. One major barrier is low educational attainment among caregivers, which may limit their understanding of treatment regimens, the long-term benefits of hydroxyurea, and the importance of consistent use [19,20]. This can affect their ability to follow dosage schedules or recognize the necessity of routine monitoring. Unemployment and poverty further constrain adherence by reducing the caregivers’ financial capacity to afford transportation to clinics, cover out-of-pocket costs for laboratory tests, or purchase hydroxyurea when not covered by insurance [21,22]. Even when hydroxyurea is provided through government programs or donations, indirect costs—such as time off work or caregiving burdens—can hinder regular follow-ups.

Additionally, psychological and cultural factors play a role. Fear of side effects, misinformation about hydroxyurea, and distrust in the healthcare system have been reported in similar settings as key deterrents to sustained use [10,11,23]. Some caregivers may also discontinue treatment due to the absence of immediate visible benefits despite its long-term impact. Forgetfulness and daily life disruptions are commonly cited practical obstacles, especially among caregivers managing multiple responsibilities. Health system limitations—such as stock-outs, long clinic wait times, and inadequate counseling from overburdened providers—compound these challenges. These systemic barriers diminish confidence in the continuity of care, thereby undermining adherence motivation.

### 3.2. Factors Associated with Adherence

Age of Children: Children aged 1–10 years had better adherence compared to those aged 11–17 years. Younger children rely more on caregivers for medication management, facilitating better adherence [23,24]. As children gain independence in adolescence, adherence often decreases [23]. Parental involvement remains crucial for maintaining high adherence levels [23].

### 3.3. Education of Caregivers

Caregivers with secondary or higher education had significantly better adherence compared to those with only primary education. Higher education levels empower caregivers with a better understanding and management of hydroxyurea treatment, enhancing communication with healthcare providers [16,25].

### 3.4. Knowledge of Hydroxyurea

Participants with good knowledge of hydroxyurea had significantly better adherence. Education and counseling improve adherence by empowering caregivers with necessary information [26]. Patient-centered care involving patients and caregivers in treatment decisions enhances adherence [27].

### 3.5. Availability of Hydroxyurea

Consistent availability of hydroxyurea at clinics significantly improved adherence. A reliable medication supply fosters trust in the healthcare system and encourages ongoing adherence [21,28,29].

## 4. Conclusions

Improving adherence to hydroxyurea in children with sickle cell anemia requires addressing educational, socioeconomic, and health literacy factors. Focused interventions can improve health outcomes in this population.

### 4.1. Study Limitations

The study’s single-center, cross-sectional design limits generalizability. Future studies should include diverse settings and use longitudinal designs. Self-reported adherence may be biased; objective measures could enhance accuracy. The study did not explore cultural factors affecting adherence, which should be addressed in future research.

### 4.2. Recommendations

Education and Counseling:

To improve hydroxyurea adherence, strategies should focus on education, access, and support. Targeted caregiver education and counseling can address misconceptions and enhance understanding of hydroxyurea’s long-term benefits. Strengthening drug supply chains and expanding health insurance coverage can reduce access barriers.

Additionally, reminder tools (e.g., phone messages), peer support groups, and integrating hydroxyurea counseling into routine clinic visits may help caregivers stay consistent with treatment. School-based awareness and community sensitization efforts could also play a key role in supporting adherence among older children and adolescents.

Age-Specific Interventions: Tailor interventions to support adolescents’ adherence.

Ensure Medication Availability: Maintain a consistent supply of hydroxyurea at clinics to support adherence.

### 4.3. Further Research

More research is needed on interventions to improve hydroxyurea adherence in children with sickle cell anemia, including educational, psychosocial, and caregiver training programs.

## Figures and Tables

**Figure 1 ijerph-22-00616-f001:**
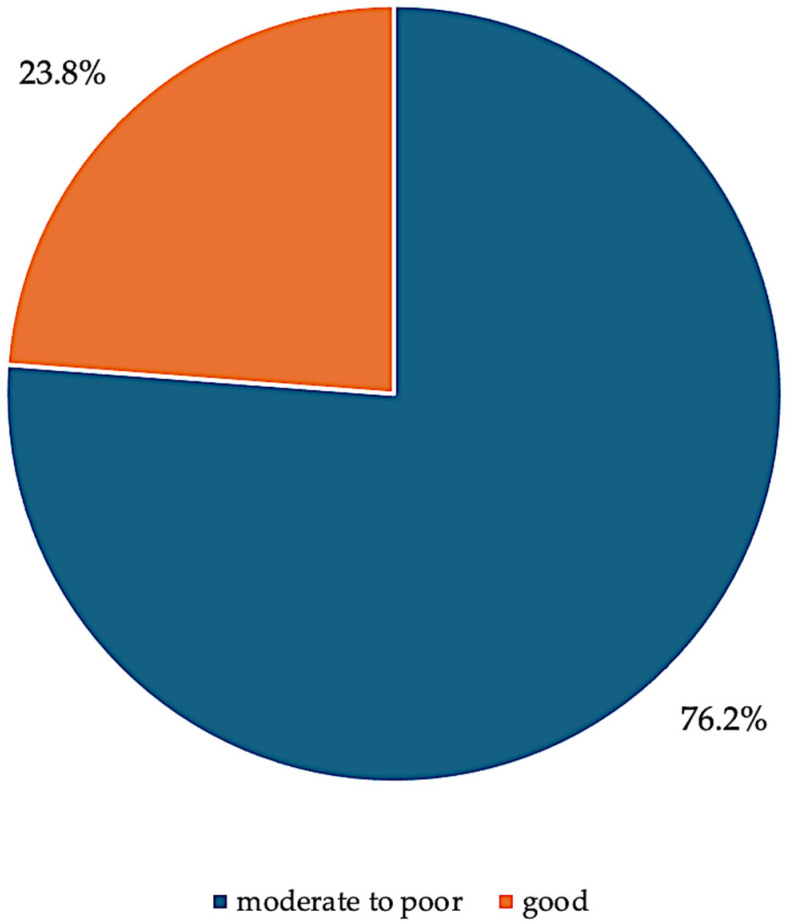
Pie chart showing adherence to Hydroxyurea treatment.

**Table 1 ijerph-22-00616-t001:** Background characteristics of the study participants (N = 172).

Variables	n	%
Age caregiver (years)		
Median (±SD)	33 (29–40)	
Child age (years)		
1–10	118	68.6
11–17	54	31.4
Median (±SD)	8 (5–12)	
Education level		
Illiterate to primary	61	35.5
Secondary	66	38.4
College/university	45	26.1
Occupation		
Employed	118	31.4
Not employed	54	68.6
Knowledge of hydroxyurea		
Good	126	73.3
Moderate	31	18.0
Poor	15	8.7
Number of family members using hydroxyurea		
One	143	85.1
More than one	25	14.9
Way to obtain hydroxyurea		
Health insurance	141	82.0
Cash/donation	31	18.0
Believe hydroxyurea reduces pain		
Yes	154	89.5
No	18	10.5
Child has health insurance		
Yes	122	70.9
No	50	29.1
Was hydroxyurea available when attending clinic?		
Yes	136	79.1
No	36	20.9
Referred		
Yes	29	16.9
No	143	83.1

**Table 2 ijerph-22-00616-t002:** Unadjusted analysis on the adherence to hydroxyurea treatment and the associated factors among study participants.

Variables	cPR 95% CI	*p*-Values
Age categories of caregiver (years)		
21–34	Ref	
35–45	0.73 (0.37, 1.42)	0.35
46–62	1.49 (0.77, 2.87)	0.23
Child age (years)		
1–10	3.29 (1.36, 7.95)	0.01
11–17	Ref	
Education level of caregiver		
Illiterate to primary	Ref	
Secondary	1.74 (0.84, 3.63)	0.13
College/university	2.26 (1.08, 4.70)	0.03
Occupation of caregiver		
Employed	0.88 (0.50, 1.55)	0.66
Not employed	Ref	
Knowledge of hydroxyurea		
Good	1.96 (1.52, 7.40)	0.02
Moderate	1.45 (0.33, 6.38)	0.62
Poor	Ref	
Number of family members using hydroxyurea		
One	Ref	
More than one	1.01 (0.47, 2.16)	0.98
Way to obtain hydroxyurea		
Health insurance	2.78 (0.91, 8.47)	0.07
Cash/donation	Ref	
Believe hydroxyurea reduces pain		
Yes	0.84 (0.38, 1.87)	0.67
No	Ref	
Child has health insurance		
Yes	1.69 (0.84, 3.41)	0.14
No	Ref	
Was hydroxyurea available when attending clinic?		
Yes	1.91 (0.80, 4.52)	0.13
No	Ref	
Referred		
Yes	0.39 (0.13, 1.18)	0.09
No	Ref	

**Table 3 ijerph-22-00616-t003:** Adjusted analysis on the adherence to hydroxyurea treatment and the associated factors among study participants.

Variables	aPR 95% CI	*p*-Value
Age categories of caregiver (years)		
21–34	Ref	
35–45	0.70 (0.33, 1.50)	0.36
46–62	2.08 (1.08, 3.97)	0.03
Child age (years)		
1–10	2.98 (1.18, 7.47)	0.02
11–17	Ref	
Education level of caregivers		
Illiterate to primary	Ref	
Secondary	1.41 (1.19, 2.87)	0.04
College/university	1.92 (1.09, 4.63)	0.01
Occupation		
Employed	1.19 (0.65, 2.19)	0.560
Not employed	Ref	
Knowledge of hydroxyurea		
Good	1.55 (1.10, 4.78)	0.04
Moderate	1.44 (1.04, 4.66)	0.01
Poor	Ref	
Number of family members using hydroxyurea		
One	Ref	
More than one	1.47 (0.67, 3.21)	0.33
Way to obtain hydroxyurea		
Health insurance	2.10 (0.58, 7.60)	0.25
Cash/donation	Ref	
Believe hydroxyurea reduces pain		
Yes	0.68 (0.27, 1.70)	0.41
No	Ref	
Child has health insurance		
Yes	0.96 (0.43, 2.12)	0.91
No	Ref	
Was hydroxyurea available when attending clinic?		
Yes	1.49 (1.03, 3.52)	0.03
No	Ref	
Referred		
Yes	0.41 (0.15, 1.14)	0.09
No	Ref	

## Data Availability

Data supporting the findings of this study are available from the corresponding author upon reasonable request.

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
