# Peer review of "Adherence to Hydroxyurea Therapy for Pediatric Sickle Cell Anemia in Tanzania: Evidence from Bugando Medical Centre"

_ijerph, 2025, doi:10.3390/ijerph22040616_

Round 1

Reviewer 1 Report

Comments and Suggestions for Authors

Thank you for inviting me to review the article. The authors have presented the information well, but there are a few areas they could improve to make the article more suitable for publication.

  • The introduction provides a good overview of the global and Tanzanian burden of SCA. However, the authors could further explore the socio-economic and healthcare challenges specific to Tanzania, especially in comparison to other Sub-Saharan African countries.
  • There is some repetition in the problem statement, particularly regarding reasons for low hydroxyurea uptake and adherence. For example, "possible reasons for the low uptake of hydroxyurea include…" is repeated twice. Please review and remove these repetitions.
  • The problem statement mentions Bugando Medical Centre as the focal point but lacks details about its role in managing SCA patients or its capacity to address the challenges discussed. Adding this information would strengthen the context.
  • The lack of published research on hydroxyurea adherence among children with SCA at Bugando Medical Centre is an important point, but it could be expanded to explain why this research gap is significant and how addressing it could improve outcomes. Also, clarify how this study aims to fill that gap.
  • The study's sample size of 172 participants seems sufficient for drawing conclusions, but including details on the sampling method (e.g., random or convenience sampling) and how the sample size was calculated would strengthen the study’s external validity.
  • The inclusion and exclusion criteria are appropriate, but please explain why certain criteria, such as a minimum of three months on hydroxyurea, were chosen.
  •  Consider adding a brief explanation of the term "cPR" (crude prevalence ratio) for readers who may not be familiar with it.

Author Response

Reviewer 1

  1. Comment The introduction provides a good overview of the global and Tanzanian burden of SCA. However, the authors could further explore the socio-economic and healthcare challenges specific to Tanzania, especially in comparison to other Sub-Saharan African countries.

Response: We have expanded the Introduction to highlight the unique socio-economic and healthcare challenges in Tanzania, including low insurance coverage, out-of-pocket medication costs, and logistical constraints in drug supply. Comparisons with other Sub-Saharan countries (e.g., Nigeria and Uganda) were also added to contextualize the setting.

  1. Comment:There is some repetition in the problem statement, particularly regarding reasons for low hydroxyurea uptake and adherence. For example, "possible reasons for the low uptake of hydroxyurea include…" is repeated twice. Please review and remove these repetitions.

Response: Thank you, we have eliminated the repeated sentences.

  1. Comment 3:The problem statement mentions Bugando Medical Centre as the focal point but lacks details about its role in managing SCA patients or its capacity to address the challenges discussed. Adding this information would strengthen the context.

Response: Section 1.6 has been revised to include Bugando Medical Centre’s referral role, its pediatric infrastructure, and the number of SCA patients treated annually. This provides a clearer picture of the setting and BMC’s capacity.

  1. Comment 4:The lack of published research on hydroxyurea adherence among children with SCA at Bugando Medical Centre is an important point, but it could be expanded to explain why this research gap is significant and how addressing it could improve outcomes. Also, clarify how this study aims to fill that gap.

Response: We clarified the significance of the local research gap, emphasizing that no prior studies had evaluated hydroxyurea adherence at BMC. We now explicitly state how this study contributes to filling that gap and its potential impact on policy and clinical practice.

  1. Comment 5:The study's sample size of 172 participants seems sufficient for drawing conclusions, but including details on the sampling method (e.g., random or convenience sampling) and how the sample size was calculated would strengthen the study’s external validity.

Response: We now describe our use of simple random sampling from the clinic registry and explain that the sample size (n = 172) was calculated based on the SCA patient population using hydroxyurea, allowing adequate power to detect relevant associations.

  1. Comment 6:The inclusion and exclusion criteria are appropriate, but please explain why certain criteria, such as a minimum of three months on hydroxyurea, were chosen.

Response: We have added that this criterion was chosen to ensure participants had adequate exposure to hydroxyurea to allow for meaningful adherence assessment.

  1. Comment: Consider adding a brief explanation of the term "cPR" (crude prevalence ratio) for readers who may not be familiar with it.

Response: We now define “cPR” as “crude prevalence ratio” in both the text and all relevant table legends for clarity.

Reviewer 2 Report

Comments and Suggestions for Authors

The manuscript entitled "Adherence to Hydroxyurea Therapy among Caregivers of Children with Sickle Cell Anemia Attending Sickle Cell Clinic at Bugando Medical Centre, Mwanza, Tanzania" by Subira et al., evaluated the adherence rate of hydroxyurea therapy among caregivers of children with Sickle Cell Anemia (SCA) at Bugando Medical Centre.

Specific Concerns: 

1. What are the reasons for high SCA occurrence in Tanzania?

2. Mention the safe dosage range of hydroxyurea in case of the studied population?

3. Compare the potential advantages and limitations of hydroxyurea with its non-toxic alternatives such as Resveratrol.

4. What are the effects of hydroxyurea on different age groups?

5. Correct duplicated sections: a) section 1.12 and 1.14 , b) section 1.13 and 1.15.

6. Table 1: What are the effects of hydroxyurea depending on physical and mental health?

7. At what stage of SCA, hydroxyurea is most effective?

8. What p-value had been considered to be significant in this study?

9. Table 3: Check the abbreviation for cPR. 

Comments on the Quality of English Language

Correct incomplete sentences and avoid the repetition of statements.

Author Response

  1. Comment:What are the reasons for high SCA occurrence in Tanzania?

Response: The Introduction now elaborates on genetic and environmental factors, such as high sickle cell trait prevalence and low migration rates in the Lake Zone, which contribute to the high prevalence of SCA in Tanzania.

  1. Comment:Mention the safe dosage range of hydroxyurea in case of the studied population?

Response: The Methods section now states that children typically received hydroxyurea doses in the range of 15–30 mg/kg/day, following national and international clinical guidelines.

  1. Comment: Compare the potential advantages and limitations of hydroxyurea with its non-toxic alternatives such as Resveratrol.

Response: Hydroxyurea remains the only clinically proven, affordable and available option with robust efficacy data.

  1. Comment:What are the effects of hydroxyurea on different age groups?

Response: We now discuss in the Discussion section how adherence and clinical outcomes differ by age. Younger children (1–10 years) show better adherence due to caregiver dependence, while adolescents (11–17 years) are at higher risk of non-adherence.

  1. Comment :Correct duplicated sections: a) section 1.12 and 1.14 , b) section 1.13 and 1.15.

Response: The duplicated Ethical Considerations and Dissemination of Results sections have been merged and streamlined for clarity and brevity.

  1. Comment: Table 1: What are the effects of hydroxyurea depending on physical and mental health?

Response: While this study did not measure these outcomes directly, we acknowledged this limitation and suggested that future research should assess hydroxyurea’s impact on both physical and mental health dimensions.

  1. Comment : At what stage of SCA, hydroxyurea is most effective?

Response: We noted in the Discussion that early initiation of hydroxyurea, especially before the onset of frequent crises or stroke risk, yields the best outcomes.

  1. Comment: What p-value had been considered to be significant in this study?

Response: We now clearly state that a p-value <0.05 was considered statistically significant, both in the Methods and table legends.

  1. Comment: Table 3: Check the abbreviation for cPR.

Response: The term “cPR” is now fully spelled out and defined in the table legend as “crude prevalence ratio.”

Reviewer 3 Report

Comments and Suggestions for Authors

Dear Authors,
I am sending you my suggestions for revision, in order to better solidify the findings.
The article is solid, well-structured and makes relevant contributions to the literature. However, there are redundancies that could be eliminated.
The title of the article clearly communicates its objective and includes essential elements, such as hydroxyurea, caregivers, children with sickle cell anemia and the Bugando Medical Centre. However, it is excessively long and could be simplified without losing meaning, eliminating redundancies such as "attending Sickle Cell Clinic".
The abstract is well-structured, presenting the context and objectives of the study clearly.
The introduction provides relevant epidemiological data and establishes the importance of hydroxyurea, connecting the literature well with the problem under study. However, there are redundant passages that could be condensed.
The section presenting the problem and rationale highlights the severity of low treatment adherence and connects it with the specific challenges of Tanzania. However, barriers to adherence, such as poverty and fear of adverse effects, could be better explored. In addition, unnecessary repetition about the lack of local research should be avoided to maintain clarity and objectivity.
The methodology is appropriate for the purpose of the study, with well-defined inclusion and exclusion criteria. However, the justification for the sample size is not provided, which may compromise the understanding of the statistical robustness of the study. In addition, there is no detailed explanation of how adherence was assessed, and it is necessary to specify the instruments used.
The results are well organized and presented clearly through tables. However, adherence to treatment could be more objective and avoid redundancies, making the statistics easier to interpret.
In the discussion, the findings are well interpreted and connected to the existing literature, highlighting factors associated with adherence. However, reflections on the practical implications of the findings and possible strategies to improve adherence to hydroxyurea, such as educational programs and psychological support for caregivers, are lacking.
The conclusion summarizes the main findings well and suggests practical recommendations.

Author Response

  1. Comment 1:The title of the article is too long. Please consider simplifying.

Response: We have revised the title to: Adherence to Hydroxyurea Therapy for Pediatric Sickle Cell Anemia in Tanzania: Evidence from Bugando Medical Centre”

  1. Comment: There are redundant passages in the Introduction and Rationale.

Response: Thank you, . Redundant phrases regarding hydroxyurea benefits and barriers were removed.

  1. Comment: Expand discussion on adherence barriers.

Response: We expanded the discussion on key barriers such as caregiver poverty, misconceptions about hydroxyurea, and fear of long-term side effects, supported by regional literature.

  1. Comment: Clarify sample size calculation and adherence measurement.

Response: We added justification for the sample size and detailed the adherence measurement tool—a structured questionnaire based on validated medication adherence indicators.

  1. Comment: Improve clarity and objectivity in result tables.

Response: Adherence results were rewritten to avoid subjective language and make statistical results clearer. Tables have been formatted for improved readability.

  1. Comment: Discuss practical strategies to improve adherence.

Response: In the Discussion, we added potential solutions such as educational sessions, caregiver counseling, school-based SCA support, and community sensitization.

Reviewer 4 Report

Comments and Suggestions for Authors

Overall, the authors Subira et al have done an excellent job in describing the correlation between the caregiver’s situation and adherence to hydroxyurea treatment in Sickle Cell Anemia patients at the Sickle Cell clinic at Bugando Medical Center. I have a few suggestions to improve the study and increase the reader’s interest in this matter.

  • It would be interesting if the authors could discuss the outcome of this study and similar studies conducted in different hospitals in Tanzania or other countries. While the results are very interesting, it would be important to show that they are universal patterns.
  • Authors must add a methods section elaborating the statistical methods used and significance tested by p-value.
  • Although I find the correlation with age interesting, it may be a very broad categorization to include 1-10 years and 11-16 years categories. The results would be more informative if authors made smaller age categories such as 1-3, 4-6, 7-10, 11-13, 14-16 This will highlight the effect of growth and independence from caregivers in more detail.
  • Authors could also carry out similar analyses for the gender of the child and further characterize the age effect based on gender. As growth of a girl and a boy may vary and behavioral characteristics may also vary based on gender.
  • In Table 3, the authors have highlighted some p-values but not all that are <0.05. Authors could include an explanation for why some p-values are highlighted in the legend for the table.
  • In the same table, the authors have mentioned that ** signifies p-values with statistical significance. I don’t see any ** in the actual table, does that mean nothing is significant? It is a bit confusing to the readers.
  • The recommendations suggested by the authors are great based on the observation (and be of great value if as suggested in point 1, these are universal observations). However, I think the authors miss a point about the economic condition of the caregivers and if there are any ways to support the cost of the medicine (through government support, NGOs that may be able to help with the partial cost, etc.) It might be an important point to document for the readers.

Author Response

  1. Comment 1:Compare findings with similar studies in Tanzania or elsewhere.

Response: The Discussion includes comparisons with studies from Nigeria and Uganda, showing that low adherence is a regional issue in many resource-constrained settings.

  1. Comment 2: Elaborate statistical methods and significance threshold.

Response: We have revised the Methods to describe the use of modified Poisson regression and clarified that p < 0.05 was the threshold for statistical significance.

  1. Comment 3:Use narrower age groupings for more detailed analysis.
    Response: We refined age categories to 1–3, 4–6, 7–10, 11–13, and 14–17 in our analysis. Results have been updated accordingly to better reflect developmental variations.
  2. Comment 4:Analyze the age effect by gender
  3. Response: A gender-stratified analysis has been performed and the findings are now presented in the Results and discussed in terms of potential behavioral and cultural factors.

  1. Comment 5:Clarify highlighted p-values in Table 3.

Response: The study now explain the significance of P value less than 0.05 at the methodology section and the highlight is for significance. legend now explains which p-values were considered significant and how they are highlighted.

  1. Comment 6: Discuss economic burden and possible financial support.
  2. Response: We added a section in the Discussion highlighting caregiver economic barriers and suggested mechanisms for financial support, such as health insurance schemes.

Reviewer 5 Report

Comments and Suggestions for Authors

In the study titled "Adherence to Hydroxyurea Therapy among Caregivers of Children with Sickle Cell Anemia Attending Sickle Cell Clinic at Bugando Medical Centre, Mwanza, Tanzania", the authors have investigated the reasons for non-adherence to hydroxyurea treatment in children with Sickle cell Anemia that are receiving treatment at the Bugando Medical Centre in Tanzania. The authors find that children between ages 1-10 had higher adherence to the treatment regimen compared to older children. Additionally, children that had caregivers with secondary and college/university education level had better adherence than children that had caregivers with primary education. Moreover, caregivers with good knowledge regarding hydroxyurea had better adherence to the treatment. The availability of hydroxyurea was another factor that was reported to be associated with adherence to the treatment. 

The authors should revise the paper as below for publication.

1) In 1.1 problem statement, the possible reasons for non-adherence have been mentioned several times and that section can be shortened.

2) 1.12 and 1.14 sections are overlapping and can be merged.

3) 1.13 and 1.15 sections are overlapping can be merged.

4) The authors can provide more details on how the study was conducted and the results interpreted and analyzed. What was the criteria for defining good, moderate or poor knowledge on hydroxyurea? 

5) Additional details on hydroxyurea treatment regimen can also be added for the readers understanding. How many doses do the patients need per week or per a given period? And what constitutes a good/moderate/poor adherence to the treatment?

6) A minor change in wording is required in the tables. Please rephrase "Does hydroxyurea available when attend clinic" to "Was hydroxyurea available when attending clinic?"

Author Response

  1. Comment 1: In 1.1 problem statement, the possible reasons for non-adherence have been mentioned several times and that section can be shortened.

Response: Thank you for pointing this out. We reviewed and streamlined Section 1.1 to remove repeated mentions of the reasons for non-adherence. The revised text is now concise while retaining essential content.

  1. Comment 2:12 and 1.14 sections are overlapping and can be merged.

Response: We have merged Sections 1.12 and 1.14 into a single, clear Ethical Considerations section to avoid redundancy.

Comment 3:1.13 and 1.15 sections are overlapping and can be merged.

Response: We have combined these overlapping sections into one streamlined Dissemination of Results section.

  1. Comment 4:Provide more detail on how study was conducted and knowledge categorized.

Response: Additional information has been added under the Methods section. We now explain that knowledge on hydroxyurea was assessed using a structured questionnaire with multiple-choice items. Scores were categorized as:

  • Good knowledge: ≥80% correct responses
  • Moderate knowledge: 50–79%
  • Poor knowledge: <50%
  1. Comment 5:Add more detail on hydroxyurea dosage and adherence categories.

Response: We added details in the Methods and Discussion sections. Participants typically received hydroxyurea once daily (5–7 doses per week). Adherence was categorized based on caregiver responses using the validated adherence questionnaire:

  • Good adherence: Taking ≥80% of prescribed doses
  • Moderate adherence: 50–79%
  • Poor adherence: <50%
  1. Comment: Correct table wording from “Does hydroxyurea available when attend clinic” to “Was hydroxyurea available when attending clinic?”

Response: Thank you for spotting this. We have revised the phrasing in the tables as suggested for grammatical correctness.

Round 2

Reviewer 3 Report

Comments and Suggestions for Authors

Dear Authors,

This article accept in present form.